# The Associations between Metalloestrogens, GSTP1, and SLC11A2 Polymorphism and the Risk of Endometrial Cancer

**DOI:** 10.3390/nu14153079

**Published:** 2022-07-27

**Authors:** Kaja Michalczyk, Patrycja Kapczuk, Grzegorz Witczak, Mateusz Bosiacki, Mateusz Kurzawski, Dariusz Chlubek, Aneta Cymbaluk-Płoska

**Affiliations:** 1Department of Gynecological Surgery and Gynecological Oncology of Adults and Adolescents, Pomeranian Medical University, 70-111 Szczecin, Poland; grzegorzwitczak9@gmail.com (G.W.); anetac@data.pl (A.C.-P.); 2Department of Biochemistry and Medical Chemistry, Pomeranian Medical University, Powstańców Wielkopolskich. 72, 70-111 Szczecin, Poland; patrycja.kapczuk@pum.edu.pl (P.K.); dchlubek@pum.edu.pl (D.C.); 3Department of Functional Diagnostics and Physical Medicine, Pomeranian Medical University in Szczecin, 70-111 Szczecin, Poland; bosiacki.m@pum.edu.pl; 4Department of Experimental and Clinical Pharmacology, Pomeranian Medical University in Szczecin, Powstańców Wielkopolskich 72, 70-111 Szczecin, Poland; mateusz.kurzawski@pum.edu.pl

**Keywords:** metalloestrogens, nutrients, trace elements, GSTP1, SLC11A2, endometrial cancer

## Abstract

Background: The incidence of endometrial cancer (EC) is still rising. Numerous risk factors including patient characteristics and molecular instability have been identified for EC. The presence of specific molecular markers allows specific diagnostic and prognostic approaches. Several single nucleotide polymorphisms (SNPs) have been identified to influence endometrial cancer risk. Metalloestrogens are metal ions which can mimic estrogen activity; however, their role in uterine pathologies remains unknown. This study aimed to investigate total blood trace elements levels and evaluate the distribution of selected genotypes in GSTP1 and SLC11A2 genes. Methods: This retrospective case-control analysis was carried out in peripheral blood samples of 110 women with endometrial cancer (EC; *n* = 21), uterine fibroma (*n* = 25), endometrial polyp (*n* = 48), and normal endometrium (*n* = 16). Analysis included measurement of metals and phosphor in serum, and of genetic polymorphisms in GST (rs1695) and SLC11A2 (rs224589) in DNA from white blood cells. Serum trace elements were measured using ICP-OES spectrometry. SNPs were identified using Taq Man real-time PCR genotyping assays. Results: The study confirmed higher age (OR 2.19, 95% CI 1.69–2.24), post-menopausal status (OR 1.89, 95% CI 1.36–1.94), and diabetes type 2 (OR 1.54; 95% CI 0.97–1.72) as independent risk factors for EC. We also found a high level of Cd (OR 1.49; 95% CI 1.31–1.63) and a low level of Co (OR 0.76; 95% CI 0.53–0.59) to be independent risk factors of EC. None of the tested polymorphisms of GSTP1 and SLC11A2 were associated with EC risk. However, high Cd (OR 1.21, 95% CI 1.15–1.29) and Ni (OR 1.07, 95% CI 1.05–1.18) serum levels were significantly associated with a SLC1A2 TG genotype, and high Cd levels with GSTP1 (OR 1.05, 95% CI 1.01–1.13).

## 1. Introduction

Endometrial cancer (EC) is the world’s most common gynecological malignancy and one of a few cancers with an increasing incidence [1]. Numerous factors have been identified to increase the risk of endometrial cancer, including patients’ age, obesity, late menopause onset, and unbalanced estrogen [2,3]. The new molecular classification, based on the Cancer Genome Atlas Research Network, distinguishes four types of EC by their molecular characteristics: Polymerase Epsilon Mutation (POLE) ultramutated, microsatellite instability hypermutated (MSI), copy-number low (CNL), and copy-number high (CNH) [4]. The presence of specific molecular markers allows new, more specific diagnostic and prognostic possibilities and the use of targeted therapy. The most common alternations were found to occur in TP53, PTEN, PI3KCA, CTNNB1, ARID1A, and KRAS pathways [4]. Multiple studies have investigated the associations between diverse single nucleotide polymorphisms (SNPs) and endometrial cancer risk. Strong associations were found for HNF1B, KLF, EIF2AK, CYP19A1, SOX4, and MYC [5].

Endometrial hyperplasia is a term describing an excessive proliferation of endometrial cells. It is a benign condition; however, it is considered as a precursor state for endometrial carcinomas [6]. Endometrial hyperplasia is usually correlated with unopposed estrogen, which in the absence of abnormal endometrial proliferation in the progesterone, stimulates abnormal endometrial proliferation [7]. Additionally, other benign uterine lesions, including uterine fibroids, were found to be hormone-dependent and estrogen is considered to be the major mitogenic factor in the uterus [8].

There is limited information on the influence of trace elements on human endometrium and their role in carcinogenesis. Metals including Zn, Cu, and Fe are essential chemicals for human well-being in trace amounts. However, when in excess, they may cause adverse effects including the induction of cellular damage, alternation in cellular homeostasis, inflammation, production of ROS (reactive oxygen species), and finally carcinogenic activity [9,10]. Additionally, certain metals such as Aluminum, Cadmium, Copper, Cobalt, Nickel, Lead, Tin, and Chromium have been found to have the ability to mimic estrogen activity and to activate estrogen receptors and were therefore named metalloestrogens [11,12,13]. This is why we decided to measure serum trace elements concentration in patients diagnosed with malignant and benign uterine lesions.

Glutathione S-transferase (GST) is an enzyme that catalyzes the conjugation of glutathione into electrophilic compounds. The enzyme not only detoxifies endogenous and exogenous species but also participates in the activation of oxidative metabolites participating in carcinogenesis, including ROS, and regulates stress-induced signaling pathways [14,15]. Some studies have presented an association between GSTP1 gene polymorphisms and increased risk of cancers, including endometrial cancer [16,17,18]. A similar association was found for endometrial hyperplasia [19]. 

SLC11 is a family of integral membrane proteins that are divalent metal ions transporters that use H^+^ -electrochemical gradient as a driving force to transport metal ions [20,21]. SLC11A2, also known as DMT1, and Nramp2, is widely distributed and expressed in the duodenum, erythroid cells, kidney, lung, brain, testis, thymus, and placenta [22]. The predominant substrates of SLC11A2 are Fe^2+^, Cd^2+^, Co^2+^, Cu^1+^, Mn^2+^, Ni^2+^, Pb^2+^, and Zn^2+^ [22]. 

SLC11A2 has an important role in iron homeostasis and transport. Mutations in the SLC11A2 gene were found in patients suffering from hypochromic microcytic anemia with serum and liver iron overload [23,24], while its activation was found to lead to severe pathologies including autophagy and cell death in Parkinson’s disease [25]. Additionally, overexpression of SLC11A2 was found to be associated with several cancers including esophageal, colorectal, and breast carcinomas [22,26,27]. They also correlated with an invasive form of the disease.

Single nucleotide polymorphism (SNP) are forms of DNA variation among individuals either caused by nucleotide transition or transversion. They may change the encoded amino acids into nonsynonymous, replacing one nucleotide with a different one, and can be silent (synonymous) or occur in the noncoding region. SNPs may result in gene expression changes, mRNA stability and protein coding. The identification of gene variations and their effect analysis may allow a betting understanding of their impact on gene function as they may be responsible for characteristics causing population diversity, genome evolution, familial or interindividual traits, and differences in disease prevalence and treatment response.

Genes and the genetic polymorphism of genes involved in metalloestrogen homeostasis (i.e., glutathione S-transferase P1 gene (GSTP1) and metal ions transport (i.e., the Solute Carrier 11 group A member 2 gene (SLC11A2) may be closely related to estrogen overstimulation and therefore serve as a potential risk factor of endometrial cancer. Genetic variation may explain the heterogeneity of patients and help identify those more susceptible to metalloestrogen stimulation. GSTP1 rs1695 and SLC11A2 rs224589 are widely studied polymorphisms of the genes coding regions of the mentioned enzymes that may alter enzyme activity among different genotypes.

This study aimed to investigate the association of the selected polymorphisms: rs1695 in GSTP1 and rs224589 in SLC11A2, together with serum and blood trace elements in different endometrial pathologies.

## 2. Materials and Methods

### 2.1. Study Participants

The study included 140 patients consecutively admitted to the Department of Gynecological Surgery and Gynecological Oncology of Adults and Adolescents, Pomeranian Medical University. This case-control study included patients with a confirmed diagnosis of endometrial cancer based on histopathological evaluation who were admitted for radical surgery. The control group consisted of patients admitted for hysteroscopy or laparoscopy/laparotomy with histopathologically confirmed benign uterine conditions or normal endometrium. The exclusion criteria included recurrence of endometrial cancer, previous cancer treatment or other types of primary care, and presence of unbalanced/untreated chronic diseases. In addition, patients with lost or incomplete data were removed from the study group. Finally, a total of 110 patients were included in the study analysis. The research was conducted in accordance with the Helsinki Declaration and with the consent of the Ethics Committee of Pomeranian Medical University in Szczecin under the number KB-0012/27/2020 on 9 March 2020. Patient characteristics are demonstrated in Table 1.

### 2.2. Laboratory Analyses

From each patient, two peripheral blood samples were collected for the study purpose: one was used to obtain serum for serum trace elements analysis, while the other was used for blood sample trace elements analysis and isolation of genomic DNA. The specimens were stored at a temperature of −80 degrees Celsius. The samples were obtained at the time of hospital admission for hysteroscopy/laparoscopy or laparotomy. Informed consent to participate in the study was obtained from all patients.

### 2.3. Trace Elements Analysis

For the purpose of elemental analysis, serum and whole blood samples underwent a microwave decomposition procedure using a microwave digestion system. After defrost and sample preparation, 65% HNO3 was added to the samples, which were then transferred into Teflon vessels and placed in the microwave. The process of sample digestion was composed of two stages: an initial of 15 min, at which the samples were gradually heated up to 180 °C, and the second of 20 min, at which the temperature was maintained at 180 °C. Samples were analyzed using inductively coupled plasma optical emission spectrometry (ICP-OES, ICAP 7400 Duo, Thermo Scientific, Waltham, MA, USA) equipped with a concentric nebulizer and cyclonic spray chamber to determine Zinc (Zn), Copper (Cu), Iron (Fe), Chromium (Cr), Cobalt (Co), Strontium (Sr), Phosphor (P), Magnesium (Mg), Cadmium (Cd), Nickel (Ni), and Manganese (Mn) content. The digested samples were further diluted 20-fold. For the analysis, 500 µL of yttrium was added with the final standard sample concentration at 0.5 mg/L and 1 mL of 1% Triton (Triton X-100, Sigma-Aldrich, Poland). The samples were further diluted with 0.075% HNO_3_ (Suprapur, Merck, Poland) up to the volume of 10 mL and stored in the fridge at 4–8 °C final until analysis. The calibration curve was constructed using multielement standard solutions (ICP multielement standard solution IV, IX and XVI, Merck, Kenilworth, NJ, USA).

### 2.4. Molecular Analysis

For the study purpose, genomic DNA was isolated from 0.2 mL of a whole blood sample (all with blood cells) using a commercial kit for genomic DNA isolation using Genomic Mini AX Blood 1000 Spin (A&A Biotechnology). The genotyping of the selected SNPs was performed using pre-designed Genotyping Assays (TaqMan real-time PCR genotyping assays, Thermo Fisher Scientific, Assay IDs: C___2967992_1_, C___3237198_20). The following SNPs were genotyped: GSTP1 rs1695 A > G and SLC11A2 rs224589 T > G.

### 2.5. Statistical Analysis

The analysis was conducted using Statistica 10, StataSoft, Poland The comparison of patient characteristics between the groups was performed using the U-Mann Whitney test. The associations between the tested SNPs and cancer risk were calculated by the comparison of the frequencies of selected genotypes among the EC and control group. Odds ratios (OR) and the corresponding confidence intervals (95% CI) for each SNP were calculated using univariable regression models. *p*-value < 0.05 was adopted as the statistical significance threshold.

## 3. Results

We found no differences between total blood/serum trace elements concentration between endometrial cancer patients and the control group of patients diagnosed with benign uterine conditions or normal endometrium. When analyzed separately, we found a significant difference for Cd levels between patients diagnosed with endometrial cancer vs endometrial polyps as patients with endometrial polyps tended to have lower Cd concentrations (*p* = 0.002). We also found a significant difference in Cu levels between patients diagnosed with uterine fibromas vs. endometrial polyps (*p* = 0.042). A similar trend was noticed for Cd levels. Another association was found for Fe concentration as patients with endometrial polyps showed lower Fe expression than patients with normal endometrium (*p* = 0.038). All of the associations are demonstrated in Table 2.

Our study showed no correlation between patients’ menopausal status and blood concentration of any of the investigated metalloestrogens (Table 3).

As a part of the study, we checked for any correlations between patients’ characteristics and serum trace element. Patients’ BMI and age did not influence either serum trace element concentration. Yet, we found some significant intracorrelations between selected trace elements levels. Serum Zn levels positively correlated with Cu, Fe, P, and Mg concentration. All of the correlations are presented in Table 4.

We conducted a univariate logistic regression model to assess the risk factors for endometrial cancer. We found patients’ age, BMI, menopausal status, and history of diabetes mellitus type 2 to influence the risk for endometrial cancer. However, trace elements concentration did not seem to influence the probability of cancer occurrence. The results are presented in Table 5.

However, upon multivariate analysis, Cadmium and Cobalt levels were found to be associated with endometrial cancer risk, while above median Co levels were found to correlate with a decreased the risk of EC. The results are displayed in Table 6.

None of the tested polymorphisms revealed a correlation with endometrial cancer risk (Table 7). However, we found some non-significant differences in genotypes frequencies among endometrial cancer patients and controls as the TG genotype was more frequently expressed among EC patients compared with controls (35% vs. 25.5%, respectively).

In our study, patients with high Cd (OR 1.21, 95% CI 1.15–1.29) and Ni (OR 1.07, 95% CI 1.05–1.18) serum levels were significantly associated with a SLC1A2 TG genotype, and high Cd levels with GSTP1 (OR 1.05, 95% CI 1.01–1.13). The prevalence of other genotypes did not seem to correlate with blood metalloestrogen levels (see Table 8).

## 4. Discussion

The etiology of most benign and malignant uterine pathologies is hormone-dependent. Estrogen is the major mitogenic factor in the uterus and is responsible for tissue remodeling during the menstrual cycle. However, abnormalities to the endometrial tissue including hormonal disbalance, changes in tissue microenvironment, abnormalities in cytokine or growth factors expression, or increased production of ROS can result in carcinogenesis. Metalloestrogens may mimic estrogen activity and activate estrogen receptors and thus can also be increased in various endometrial pathologies.

In the presented study, we analyzed the trace elements concentrations in different uterine pathologies. We did not find any significant correlations between the selected microelements and endometrial cancer when compared to the control group of benign uterine pathologies/normal endometrium as a whole; however, we discovered some significant correlations only when we divided the control group into separate categories for each histopathological diagnosis. We found some correlations for Cd, Cu, and Fe serum concentration. Higher Cd serum concentrations were observed in patients diagnosed with endometrial cancer when compared with patients diagnosed with endometrial polyps (*p* = 0.002). Additionally, higher Cd and Cu concentrations were found in patients diagnosed with uterine fibromas when compared with endometrial polyps; however, the median concentrations of Cd and Cu were still lower in patients diagnosed with uterine fibromas than in EC patients. Moreover, patients diagnosed with endometrial polyps also had significantly higher Fe serum expression when compared with patients with normal endometrium. Elevated copper levels (both serum and tissue) have been found in multiple cancers [28]. As copper is a co-factor in redox reactions of enzymes participating in basic biological reactions required for cell growth and development including superoxide dismutase and cytochrome c oxidase, its increased amount may correlate with an increased need for tissue proliferation. As uterine fibromas are benign tumors which are also characterized with a rapid growth, this may be the reason for an increased trace elements concentration. Further studies are required to deepen the knowledge on the role of trace elements and their distribution in different uterine pathologies.

So far, only a few studies have evaluated metalloestrogen concentrations in endometrial cancer and resulted in conflicting reports. Due to the high importance of metalloestrogens and their unexplained role in uterine pathologies, we decided to conduct a study to further evaluate their distribution in patients with different uterine conditions. Atakul et al. [29] found associations between serum Cu and Zn. In accordance with their study, lower Cu, Zn, and Cu/Zn ratio was found in patients diagnosed with endometrial cancer than in control group. Cu concentration also inversely corelated with myometrial invasion. Yaman et al. [30] investigated trace metal concentration in different cancerous and noncancerous endometrial, ovary, and cervical tissue samples. The authors found increased Fe, similar levels of Cu, and lower levels of Zn in endometrial cancer patients when compared with controls. Additionally, Rzymski et al. [31] investigated metal accumulation in uterine tissue samples. Compared with normal endometrium, endometrial cancer, hyperplasia, and CIN samples revealed significantly increased levels of Cd, Pb, Cu, Mn, and Cu/Zn ratio. Both current and former smoking status were associated with significantly higher Cd and Pb levels. Additionally, endometrial polyps, when compared with histologically normal endometrium, showed increased median concentrations of Al, Cd, Ni, and Pb. There was no significant association for Cu/Zn ratio. The study showed no correlation between patients’ age, menopausal status and the concentration of any of the investigated elements in endometrial tissue sample. Additionally, in our study, the menopausal status did not influence any of the assessed metalloestrogen levels. As there are still few data on the role of metalloestrogens and their levels in patients with different gynecological conditions, further research is needed to evaluate their significance. It would also be interesting to assess endometrium/uterine tissue metalloestrogen levels and compare their expression with serum levels.

The limitation of our study is that we did not ask the patients about any recent use of dietary supplements. As the supplements containing elements such as Zn, Cu, or Fe are widely accessible, their use might have influenced their blood concentration. However, as the supplements in Eastern Europe are still not as popular and accessible as in the western countries, we believe that their use was very limited. There is no obvious explanation for the differences observed between the studies in serum microelement concentration; however, the discrepancies may be caused based on the study size and patient characteristics. The differences may be also caused by the methods used for serum microelement analyzes, e.g., colorimetry, spectrophotometry, ICP, which each has different sensitivity and selectivity. In this study, we used ICP-OES—a very accurate method to analyze the trace elements concentration, which was also previously described in other trace elements analyses.

As a part of the study, we also conducted univariate analysis to evaluate the influence of patient characteristics and the assessed variables on the endometrial cancer risk. We confirmed that a high patient age, BMI, post-menopausal status and diabetes type 2 were significant risk factors of EC (see Table 5). Upon multivariate analysis, we found cadmium and cobalt levels to be associated with endometrial cancer risk. Higher Cd levels were found to be associated with increased endometrial cancer risk (OR 1.49, *p* = 0.0367), while above median Co levels were found to correlate with lower EC risk (OR 0.76, *p* = 0.0423). In a study by Rzymski et al. [31], the author found both current and former smoking status to be associated with significantly higher Cd and Pb levels. In our study, we did not further evaluate the correlation between cigarette use and trace element levels, as smoking was found not to be associated with endometrial cancer risk in our analysis; however, the study findings by Rzymski may be a potential explanation for this observation and require further analysis. Cobalt is an essential component of vitamin B12. Vitamin B12 is essential for the maintenance of DNA methylation, repair, synthesis, and thus for cell development and proper function. Accumulating evidence suggests the role of increased levels of folate and B-vitamins to have a role in cancer formation. Supplemental use of vitamin B12 intake was found to be associated with type 2 EC. However, the study did not include multivariable analysis. There is limited and still conflicting evidence regarding the influence of vitamin B12 and folic acid on cancer incidence. Further studies are required to explain their effect on cancer risk and cancer formation.

GSTs have a particularly important detoxification capability that protects against environmental and oxidative stress. Recent studies reported alternated GST expression to be associated with increased cancer risk [32]. As genetic polymorphisms can alternate enzyme expression, we decided to measure selected SNPs. We analyzed whether polymorphisms rs1695 in GSTP1 and rs224589 in SLC11A2 genes are associated with the risk of endometrial cancer. In our study, we found no significant association between GSTP1 and SLC11A2 polymorphisms and endometrial cancer prevalence.

A meta-analysis by Zhao et al. [33] tried to evaluate the associations between GSTP1 Ile105Val polymorphism and gynecological cancer susceptibility; however, the researchers found no significant associations with any genetic model even when accounting for cancer type, ethnicity, and smoking status. The study included a limited population as only two studies discussed endometrial cancer patients. A study by Ozerkan et al. [34] found no associations between GSTP1 polymorphism and EC risk in Caucasian population. On the other hand, a study by Chan et al. [35] revealed GSTP1 Ile(105)Val polymorphism to be associated with an increased risk of endometrial cancer. In our study, we found no associations between GTSP1 polymorphism and EC risk; however, as there are contradictory results, further research is needed.

This was the first study to analyze functional polymorphism in SLC11A2 gene in endometrial cancer patients. Previous studies described SLCA11A2 overexpression in breast carcinomas [22,26]. As endometrial cancer is also hormone dependent, we wanted to determine if its gene polymorphism affects the risk of EC. Even though we found no correlation between SLC11A2 polymorphisms and endometrial cancer risk, we found a positive correlation between SCL11A2 TT genotype and Cu concentration (*p* = 0.037). On the other hand, patients with TG genotype demonstrated lower Cd and Ni levels (*p* = 0.023 and *p* = 0.040, respectively). Further research is needed to evaluate the correlations between selected genotypes and trace elements levels due to the limited number of patients included in the analysis.

## 5. Conclusions

The study confirmed higher patient age, post-menopausal status, presence of diabetes type 2, and higher BMI as independent risk factors for endometrial cancer. Menopausal status did not influence metalloestrogen levels. High serum cadmium and low cobalt concentrations were found to influence endometrial cancer risk. None of the tested genetic polymorphisms (rs1695 in GSTP1 and rs224589 in SLC11A2) were found to be associated with endometrial cancer risk. However, GTSP1 and SLC11A2 SNPs may correlate with selected trace elements concentrations as high Cd and Ni serum levels were significantly associated with the SLC1A2 TG genotype and high Cd levels with GSTP1. Analyses of a more extensive study group should be performed to confirm our findings.

## Figures and Tables

**Table 1 nutrients-14-03079-t001:** Group characteristics.

		Number of Patients
Age	<50 years	55
	≥50–60 years	31
	≥60 years	33
BMI	<25	35
	≥25 <30	36
	≥30	25
Cigarette smoking	Yes	7
	No	101
Menopause status	Before	36
	After	64
Type 2 Diabetes	Yes	15
	No	93
Hypothyroidism	Yes	18
	No	90
Histopathological diagnosis	Endometrial cancer	21
	Uterine fibroma	25
	Endometrial polyp	48
	Normal endometrium	16

**Table 2 nutrients-14-03079-t002:** Associations between specific trace elements levels in selected groups.

		EC	Control	*p*-Value	EC	Polyp	*p*-Value	EC	Normal Endometrium	*p*-Value	EC	Fibroma	*p*-Value	Fibroma	Polyp	*p*-Value	Fibroma	Normal Endometrium	*p*-Value	Polyp	Normal Endometrium	*p*-Value
Cu	Below median	9	44	0.620	8	24	0.412	10	8	0.858	11	11	0.545	8	26	0.042	12	8	0.853	22	8	0.837
	Above median	11	42		11	21		10	9		9	13		16	18		12	9		22	9	
Zn	Below median	12	41	0.321	10	22	0.784	11	7	0.402	11	11	0.545	12	22	1.000	12	8	0.853	22	8	0.837
	Above median	8	45		9	23		9	10		9	13		12	22		12	9		22	9	
Pb	Below median	12	34	0.198	11	16	0.248	12	6	0.130	10	10	0.525	12	16	0.355	14	6	0.151	20	6	0.224
	Above median	7	39		7	20		7	10		8	12		9	20		9	10		14	9	
Cd	Below median	6	44	0.107	4	27	0.002	9	9	0.630	8	12	0.536	5	27	0.000	8	9	0.081	22	7	0.613
	Above median	12	37		14	14		11	8		9	9		19	14		15	5		21	9	
Co	Below median	5	18	0.710	5	9	0.686	4	3	0.782	5	5	0.653	6	9	0.705	5	3	0.858	9	5	0.228
	Above median	4	19		4	10		4	4		4	6		5	10		6	3		11	2	
Fe	Below median	12	41	0.321	12	20	0.281	10	8	0.858	12	10	0.226	11	23	0.612	9	11	0.086	18	12	0.038
	Above median	8	45		8	24		10	9		8	14		13	21		15	6		26	5	
P	Below median	7	45	0.205	7	25	0.106	8	10	0.254	11	11	0.545	9	25	0.128	10	10	0.279	22	8	0.837
	Above median	12	40		13	19		12	7		9	13		15	19		14	7		22	9	

**Table 3 nutrients-14-03079-t003:** Serum metalloestrogen concentration in pre- and postmenopausal patients.

Trace Element	Median Serum Concentration	N of Patients before Menopause	N of Patients after Menopause	OR	*p*-Value
Cu	0.875	34	39	1.88	0.182
Zn	0.989	34	39	0.70	0.442
Cd	0.003	9	19	1.72	0.505
Co	0.007	11	17	1.55	0.576
Fe	1.301	34	39	0.62	0.309
P	111.230	35	37	1.01	0.984

**Table 4 nutrients-14-03079-t004:** Spearman correlation.

	BMI	Age	Zn	Cu	Fe	Cr	Co	Sr	P	Mg	Cd	Ni
BMI	1.000	0.178	0.027	0.198	−0.077	0.019	0.150	0.007	−0.099	−0.008	−0.165	−0.015
Age	0.178	1.000	−0.067	0.188	−0.031	0.051	−0.183	0.056	0.033	0.022	−0.078	−0.018
Zn	0.027	−0.067	1.000	** 0.392 **	** 0.215 **	0.008	0.066	0.048	** 0.281 **	** 0.253 **	−0.062	0.009
Cu	0.198	0.188	** 0.392 **	1.000	0.169	−0.068	0.032	0.138	** 0.529 **	** 0.492 **	−0.169	0.090
Fe	−0.077	−0.031	** 0.215 **	0.169	1.000	0.146	0.053	0.117	** 0.314 **	** 0.339 **	0.024	−0.068
Cr	0.019	0.051	0.008	−0.068	0.146	1.000	0.133	0.217	−0.291	0.110	0.032	0.331
Co	0.150	−0.183	0.066	0.032	0.053	0.133	1.000	** 0.514 **	**−0.402**	−0.021	0.237	0.034
Sr	0.007	0.056	0.048	0.138	0.117	0.217	** 0.514 **	1.000	−0.024	** 0.264 **	−0.049	0.038
P	−0.099	0.033	** 0.281 **	** 0.529 **	** 0.314 **	−0.291	** −0.402 **	−0.024	1.000	** 0.546 **	−0.036	** −0.298 **
Mg	−0.008	0.022	** 0.253 **	** 0.492 **	** 0.339 **	0.110	−0.021	** 0.264 **	** 0.546 **	1.000	0.096	−0.136
Cd	−0.165	−0.078	−0.062	−0.169	0.024	0.032	0.237	−0.049	−0.036	0.096	1.000	−0.372
Ni	−0.015	−0.018	0.009	0.090	−0.068	0.331	0.034	0.038	** −0.298 **	−0.136	−0.372	1.000

The underlined variables are significant with *p* < 0.05.

**Table 5 nutrients-14-03079-t005:** Univariate logistic regression model.

		Endometrial Cancer	Control Group	OR	Upper 95%CI	*p*-Value
Age	Above median	17	39	2.21	1.86–2.29	0.001
Menopause	Yes	19	45	1.48	1.20–1.49	0.001
Grading	1	1	0	-	-	0.733
	2–3	17	2	-		
Staging	1–2	11	1	-	-	0.773
	3–4	1	0	-		
Smoking	Yes	3	4	1.71	1.49–183	0.086
BMI		20	88	1.34	1.29–1.51	0.0032
Diabetes type 2	Yes	10	5	1.26	1.14–1.30	0.000
Hashimoto	Yes	1	17	0.53	0.44–0.65	0.121
Cu	Above median	11	42	1.28	1.17–1.31	0.620
Zn	Above median	8	45	0.71	0.70–0.86	0.321
Pb	Above median	7	39	0.63	0.60–0.71	0.198
Cd	Above median	12	37	1.38	1.32–1.44	0.107
Co	Above median	4	19	0.76	0.74–0.79	0.710
P	Above median	12	40	1.62	1.57–1.66	0.205

**Table 6 nutrients-14-03079-t006:** Multivariate logistic regression model.

		Endometrial Cancer	Control Group	OR	95%CI	*p*-Value
Age	Above median	17	39	2.19	1.69–2.24	0.0028
Menopause		19	45	1.89	1.36–1.94	0.0002
Grading		1	0			ND
		17	2			
Staging		11	1			ND
		1	0			
Smoking		3	4	2.41	2.30–3.61	0.0824
BMI		20	88	1.22	1.18–1.38	0.0502
Diabetes		10	5	1.54	0.97–1.72	0.0029
Hashimoto		1	17	1.09	0.86–1.12	0.0943
Cu	Above median	11	42	1.61	1.29–1.80	0.2319
Zn	Above median	8	45	0.75	0.70–1.12	0.0540
Pb	Above median	7	39	0.66	0.59–0.66	0.2743
Cd	Above median	12	37	1.49	1.31–1.63	0.0367
Co	Above median	4	19	0.76	0.53–0.59	0.0423
P	Above median	12	40	1.22	1.16–1.23	0.2036

**Table 7 nutrients-14-03079-t007:** The associations between the analyzed SNPs and endometrial cancer risk.

	Genotype	Number of Cases	Endometrial Cancer	Control Group	OR	95%CI	*p*-Value
SLC11A2	GG	85	12 (60.0%)	73 (71.6%)	1	-	
	TG	33	7 (35.0%)	26 (25.5%)	1.638	1.235–1.702	0.348
	TT	4	1 (5.0%)	3 (2.9%)	2.027	1.649–2.094	0.464
GSTP1	AA	52	9 (45.0%)	43 (47.8%)	1	-	-
	AG	50	10 (50.0%)	40 (44.4%)	1.194	1.088–1.217	0.710
	GG	8	1 (5.0%)	7 (7.8%)	0.683	0.598–0.702	0.654

**Table 8 nutrients-14-03079-t008:** Polymorphisms prevalence with regard to metalloestrogen levels.

**Studied Population (n)**	**Cu**		***p*-Value**	**Studied Population (n)**	**Zn**		***p*-Value**	**Studied Population (n)**	**Fe**		***p*-Value**	**Studied Population (n)**	**Co**		***p*-Value**	**Studied Population (n)**	**Cd**		***p*-Value**
	**OR**	**95%CI**			**OR**	**95%CI**			**OR**	**95%CI**			**OR**	**95%CI**			**OR**	**95 %CI**	
71	-	-	-	71	-	-	-	71	-	-	-	32	-	-	-	65	-	-	-
31	1.11	1.04–1.12	0.535	31	0.98	0.84–1.04	0.282	31	1.29	1.19–1.35	0.518	12	1.15		0.361	30	1.21	1.15–1.29	0.023
4	1.09	1.02–1.11	0.037	4	0.87	0.85–0.92	0.553	4	1.34	1.30–1.41	0.722	2	1.27	1.17–1.30	0.114	4	1.20	1.19–1.24	0.091
51	-	-	-	51	-	–	-	51	-	-	-	17	-	-	-	46	-	-	-
47	1.19	1.17–1.20	0.312	47	0.79	0.76–0.86	0.222	47	1.25	1.21–1.30	0.157	24	1.23	1.20–1.31	0.154	45	1.09	1.04–1.11	0.423
8	1.15	1.11–1.16	0.484	8	0.94	0.92–0.99	0.227	8	1.29	1.22–1.35	0.327	5	1.19	1.16–1.28	0.127	8	1.05	1.01–1.13	0.032
**Studied Population (n)**	**Sr**		***p*-Value**	**Studied Population (n)**	**Cr**		***p*-Value**	**Studied Population (n)**	**Mg**		***p*-Value**	**Studied Population (n)**	**Mn**		***p*-Value**	**Studied Population (n)**	**Ni**		***p*-Value**
	**OR**	**95%CI**			**OR**	**95%CI**			**OR**	**95%CI**			**OR**	**95%CI**			**OR**	**95%CI**	
63	-	-	-	27	-	-	-	71	-	-	-	55	-	-	-	36	-	-	-
29	0.89	0.82–0.90	0.400	11	0.76	0.74–0.88	0.355	31	1.24	1.19–1.28	0.518	23	0.86	0.84–1.02	0.058	20	1.07	1.05–1.18	0.040
4	0.77	0.74–0.81	0.237	1	-	-	-	4	1.11	1.09–1.21	0.722	2	-	-	-	2	-	-	-
47	-	-	-	17	-	-	-	51	-	-	-	37	0.97	0.93–0.99	0.087	34	-	-	-
43	0.91	0.89–0.94	0.196	22	1.01	0.97–1.08	0.624	47	1.15	1.12–1.20	0.145	36	0.90	0.87–0.97	0.869	21	1.10	1.08–1.21	0.827
6	0.83	0.80–0.86	0.274	0	-	-	-	8	1.27	1.22–1.28	0.058	7	0.86	0.85–0.93	0.680	3	1.16	1.15–1.19	0.564

## Data Availability

The data presented in this study are available on request from the corresponding author. The data are not publicly available due to ethical restrictions.

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
