# Peer review of "The Associations between Metalloestrogens, GSTP1, and SLC11A2 Polymorphism and the Risk of Endometrial Cancer"

_nutrients, 2022, doi:10.3390/nu14153079_

Round 1

Reviewer 1 Report

The manuscript is well written, and data has been presented in a systematic way. I do think that there may be a room for improvements in mentioning about the methods. Also, it is better to add some thought on what kind of laboratory functional assay needs to be performed on the collected blood samples which further establish the association of metalloestrogen, GSTP1 and SLC11A2 polymorphism and how can could increase the risk of endometrial cancer. I think those addition would further enhance the merits of manuscript.

Thank you

Author Response

Dear reviewer,

We added some additional information to the methods section

We have also improved the abstract and conclusions, some of the tables and provided additional statistical analysis.

Thank you for your comments and please see the improved version of the manuscript

Best regards

Reviewer 2 Report

Referee report Michalczyk_Nutrient_5 July 2022

The associations between metalloestrogens, GSTP1, and SLC11A2 polymorphism and the risk of endometrial cancer.

Kaja Michalczyk 1*,, Patrycja Kapczuk2 , Grzegorz Witczak1 , Mateusz Bosiacki3 , Mateusz Kurzawski4 , Dariusz , Chlubek 2 and Aneta Cymbaluk-Płoska

 Summary in words of the reviewer:

1.       Estrogen overstimulation is a recognized as a risk factor of endometrial cancer (EC). Certain trace metals, and foreign metals, may in excess amount mimic estrogen activity (metalloestrogens), and may thereby be a risk factor of endometrial cancer (EC). This may be closely related to genetic polymorphism of genes involved in homeostasis, (i.e. glutathione S-transferase P1 gene (GSTP1)) and metal ions transport (i.e. the Solute Carrier 11 group A member 2 gene (SLC11A2)).
   This retrospective case-control analysis was done in peripheral blood samples of 110 women with endometrial cancer (EC; n=21), uterine fibroma (n=25), endometrial polyp (n=48) and normal endometrium (N=16). Analysis  included measurement of metals and phosphor in serum, and of genetic polymorphisms in GST (rs1695) and SLC11A2 (rs224589) [in DNA from white blood cells ??].
   This study confirmed higher age (OR 4.3, 95% CI 2.1-24.3), post-menopausal status (OR 12.0, 95% CI 2.7-66.6), and diabetes type 2 (OR 18.6; 95% CI 8.3-36.9), as independent risk factors for EC. The authors also found that a high level of Cd (OR 2.5; 95% CI ???), and a low level of Co (OR 0.44; ???)  were independent risk factors of EC.  None of the tested polymorphisms of GSTP1 and SLC11A2 were associated with EC risk.
   However, (high?) Cd and Ni serum levels are significantly associated with a SLC1A2 TG genotype, and (high?) Cd levels with GSTP1 but the study does not clearly mention how strongly and in what direction (high or low?). [Suggestion: Mention ORs and 95% CIs.]

 Introduction

2.       Line 94: unfinished sentence “together with serum and blood trace…” 

3.       Line 94: What is rs224589 ? (And in the abstract: rs169?) (Yes, I know, but the virgin reader probably doesn’t. Explain SNPs briefly in the introduction.)

 Methods

4.       Page 3, line 111, Table 1: Since the median Age is used as threshold in further analyses, it may be explicitly mentioned.  

5.       Page 3, line 119, Paragraph 2.3 Trace elements analysis . Suggestion: mention trace elements that have been analyzed; for one time fully written: copper (Cu), Phosphor (P) etc… (particularly since P is not a metal)

6.       Page 4, line 135, paragraph 2.4 Molecular analysis. I think that not many readers of Nutrients are acquainted with SNPs. A few more words about SNP analysis may be helpful, partly in the introduction (what is a SNP and what does is stand for?) and how are SNPs measured (in M&M).

7.       Page 4, line 136: From which cells was DNA extracted? I guess from all white blood cells? Or from a specific subpopulation?

Results

8.       Pages 4-5, Table 2: In my PDF-version, Table 2 came out quite skewed.

9.       Page 6, Table 4, line 171; “* The underlined variables are significant with p<0.05” Considering that this table presents 144-12 = 132 Spearman comparisons, I think that a correction for multiple testing would be more appropriate. Ask your statisticians.

10.   Page 6; Table 5, Age: Numbers do not total 110 ? Why not? (I did not check all 2x2 tables, but the authors are advised to do so).

11.   Page 6; Table 5. (and Page 7, Table 6): Does it make sense to mention both OR and inverse OR (I suggest to present the OR and 95% CIs of the higher category within a group compared to the lower category? See my personal summary.

12.   Page 6; Table 5. (and Page 7, Table 6): I miss obesity (BMI) in the uni- and multivariable analysis. If dependency of obesity and DM2 obscured (multi-variable) analysis, mention that in the text.

13.   Page 7, Table 6: The 95% CI and p-values of the ORs of Cd (OR 2.53; 95% CI .26-1.71; p=0.0472) and Co (OR 0.44; 95% CI .39-6.02; p=0.0312) make no sense to me. (The 95% CI of a significant OR is not supposed to trespass 1.0).

14.   Page 7; Table 6: Why are 75% CIs mentioned?

15.   Page 8, Table 8: see Table 4: The pdf gives a terribly distorted table. I can read that Cd and Ni serum levels are significantly associated with a SLC1A2 TG genotype, and Cd levels with GSTP1 but not how strongly and in what direction (OR and 95% CI are missing),

Discussion

16.   Page 10, lines 261-262: “We found patients’ age, menopausal status, and history of diabetes mellitus type 2 to influence the risk for endometrial cancer.” This can (and I think should) be formulated more precise: “We confirmed that a high patient age, post-menopausal status and diabetes type 2 were significant risk factors of EC.” (Here also would suit a remark about obesity or BMI).

17.   Page 10, lines 262-264: “Also, serum P (OR 1.93, p=0.205), whole blood Pb (OR 0.51, p=0.198), and Cd (OR 2.38, p=0.107) levels seemed to correlate with the risk of endometrial cancer; however, the results were insignificant.”  Both the relatively low OR (~0.5 and ~2) and absence of significance contradict this suggestion.  

18.   Page 10, lines 266-268: High Cd, low Co and EC risk: Although smoking is not a Nutrient in a strict sense, it is a well-known source of carcinogens sold over the counter. May there be an association between Cd and (former) smoking, as suggested earlier (ref Rzymski [31], lines 233-239)? And, pondering about poison and necessary trace elements in food and supplements, how can we interpret the protective association between Co and EC?

19.   However, smoking is associated with a lower risk of EC. Does that make sense?

20.   Page 10, lines 269-277. I am not sure if discussion of SOD1, SOD2 and MnSOD is directly relevant to this paper, since they were not studied. If so: explain why.

21.   Page 10, line 283 and further. This study did not find an association between GSTP1 and SLC11A2 polymorphism and EC. But then, how should we interpret the association of specific gene types with Cd serum levels (of which higher levels are, according to this study, associated with EC risk). Small numbers? Confounding?

22.   Page 11, line 305: “Serum cadmium and cobalt concentrations were found to influence endometrial cancer risk.” Be more explicit in how Cd and Co are associated with EC risk (see page 10, lines 266-268): Suggestion: ‘High Cd and low Co serum levels were associated with EC risk.’

Author Response

Dear reviewer, thank you for your comments

We have improved the abstract and the data from the statistic analysis in accordance with your suggestions, please see the abstract in the manuscript 

This study confirmed higher age (OR 2.19, 95% CI 1.69-2.24), post-menopausal status (OR 1.89, 95% CI 1.36-1.94), and diabetes type 2 (OR 1.54; 95% CI 0.97-1.72), as independent risk factors for EC. The authors also found that a high level of Cd (OR 1.49; 95% CI 1.31-1.63), and a low level of Co (OR 0.76; 95% CI 0.53-0.59)  were independent risk factors of EC.  None of the tested polymorphisms of GSTP1 and SLC11A2 were associated with EC risk.  However, high Cd (OR 1.21, 95% CI 1.15-1.29) and Ni  (OR 1.07, 95% CI 1.05-1.18) serum levels are significantly associated with a SLC1A2 TG genotype, and high Cd levels with GSTP1  (OR 1.05, 95% CI 1.01-1.13).

We have added some additional paragraphs in accordance with your kind comments. We have provided some additional information on the SNPs, why they were measured, and the methodology used. Moreover, we provided some more detailed explanation of the previously “unclearly written” parts of the manuscript 

Please see the reviewed version of the manuscript

Table 2 - the initial outlook of the table we submitted was different from the one edited by the editorial office- the table requires further graphical changes- to be discussed with the editorial team by the final submission of the manuscript; sorry for that; we also decided to use short cuts in the names of the groups to ease the reading of the table 

We improved table 5 to make it more easily to read and to understand the presented data

We were advised to put 75 CI data by our statistician as he thought it increased the credibility of the data- however, after having received the comments, we believe It is unnecessary and we decided to delete this part of table 6; please see the improved version of the table

We have also improved table 7 and 8 - please see the tables

7.       Page 4, line 136: From which cells was DNA extracted? I guess from all white blood cells? Or from a specific subpopulation?

all white blood cells- we added this information

Discussion

We improved the discussion in accordance with your suggestions; please see the improved version of the manuscript

We added some additional information about the possible correlation between cobalt and cancer and Cd and smoking

As to the comment of smoking- our study shows a positive, non significant correlation between EC risk and smoking, thus it may possibly make sense

20.   Page 10, lines 269-277. I am not sure if discussion of SOD1, SOD2 and MnSOD is directly relevant to this paper, since they were not studied. If so: explain why. - I deleted this part of the manuscript as I also believe it is unnecessary